# Comparative efficacy of prophylactic anticonvulsant drugs following traumatic brain injury: A systematic review and network meta-analysis of randomized controlled trials

**Bo-Cyuan Wang**[1,2◑], **Hsiao-Yean Chiu**[1◑], **Hui-Tzung Luh**[3,4,5], **Chia-Jou Lin**[1], **Shu-Hua Hsieh**[1,6], **Ting-Jhen Chen**[1], **Chia-Rung Wu**[1,6], **Pin-Yuan Chen**[7,8,9]*

1 School of Nursing, College of Nursing, Taipei Medical University, Taipei, Taiwan, 2 Department of Nursing, New Taipei City Municipal Tucheng Hospital (Built and Operated by Chang Gung Medical Foundation), Taipei, Taiwan, 3 Department of Neurosurgery, Taipei Medical University Shuang Ho Hospital, New Taipei City, Taiwan, 4 Taipei Neuroscience Institute, Taipei Medical University, New Taipei City, Taiwan, 5 Graduate Institute of Clinical Medicine, National Taiwan University, Taipei, Taiwan, 6 Department of Nursing, Far Eastern Memorial Hospital, New Taipei, Taiwan, 7 Department of Neurosurgery, Chang Gung Memorial Hospital, Keelung, Taiwan, 8 School of Medicine, College of Medicine, Chang-Gung University, Taoyuan, Taiwan, 9 Community Medicine Research Center, Chang Gung Memorial Hospital, Keelung, Taiwan

◑ These authors contributed equally to this work.
* pinyuanc@gmail.com

**Data Availability Statement:** All raw data files are available in the supplemental materials.

## Abstract

We systematically compared the effects of prophylactic anticonvulsant drug use in patients with traumatic brain injury. We searched four electronic databases from their inception until July 13, 2021. Two researchers independently screened, appraised, and extracted the included studies. Network meta-analysis using multivariate random effects and a frequentist framework was adopted for data analysis. The risk of bias of each study was assessed using the Cochrane risk of bias tool, and confidence in evidence was assessed through confidence in network meta-analysis (CINeMA). A total of 11 randomized controlled trials involving 2,450 participants and six different treatments (i.e., placebo, carbamazepine, phenytoin, levetiracetam, valproate, and magnesium sulfate) were included. We found that anticonvulsant drugs as a whole significantly reduced early posttraumatic seizures (PTS) but not late PTS compared with placebo (odd ratios [ORs] = 0.42 and 0.82, 95% confidence intervals [CIs] = 0.21–0.82 and 0.47–1.43). For the findings of network meta-analysis, we observed that phenytoin (ORs = 0.43 and 0.71; 95% CIs = 0.18–1.01 and 0.23–2.20), levetiracetam (ORs = 0.56 and 1.58; 95% CIs = 0.12–2.55 and 0.03–84.42), and carbamazepine (ORs = 0.29 and 0.64; 95% CIs = 0.07–1.18 and 0.08–5.28) were more likely to reduce early and late PTS compared with placebo; however, the treatment effects were not significant. Sensitivity analysis, after excluding a study enrolling only children, revealed that phenytoin had a significant effect in preventing early PTS (OR = 0.33; 95% CI = 0.14–0.78). Our findings indicate that no antiepileptic drug had an effect on early or late PTS superior to that of another; however, the sensitivity analysis revealed that phenytoin might prevent early PTS.

**Funding:** This meta-analysis was supported by grants from the Chang Gung Medical Foundation (CMRPVVK0071) and Chang Gung Memorial Hospital, Keelung branch (CRRPG2K0041)." The funders had no role in study design, data collection and analysis, decision to publish, or preparation of the manuscript.

**Competing interests:** The authors have declared that no competing interests exist.

Additional studies with large sample sizes and a rigorous design are required to obtain high-quality evidence on prophylactic anticonvulsant drug use in patients with traumatic brain injury.

## Introduction

Traumatic brain injury (TBI) is a serious public health problem worldwide. Approximately 2.8 million TBI-related emergency department visits, hospitalizations, and deaths occurred in the United States in 2013 [1]. Patients with head injuries can develop post-traumatic seizures (PTS), which vary in severity and affect up to 56% of cases [2–4], and these seizures may adversely affect the prognosis of functional outcomes [5, 6].

The Brain Trauma Foundation *Guidelines for the Management of Severe Traumatic Brain Injury (Fourth Edition)* recommend pharmacological therapy, particularly phenytoin, to reduce the incidence of early but not late PTS [7]. A pairwise meta-analysis in 2012 reported that phenytoin was not superior to levetiracetam in preventing early posttraumatic attacks [8]. Another meta-analysis published in 2015 indicated that traditional antiepileptic drugs (AEDs), namely phenytoin and carbamazepine, may reduce the risk of early PTS compared with placebo or standard care, and that AEDs were not more effective than placebo or standard care in relieving late PTS [9]. Because previous studies have produced inconsistent findings, the clinical application of these results has thus been restricted. In addition, one more randomized controlled trial (RCT) has been published since the two meta-analyses [10]. Conducting an updated and sophisticated meta-analysis to address relevant research questions is therefore of clinical relevance.

Network meta-analysis (NMA) is an advanced method that can simultaneously compare multiple treatments by estimating direct and indirect treatment effects, thereby providing physicians with more useful clinical information regarding treatment selection. In this systematic review, we used NMA to compare the efficacy of different AEDs in treating early and late PTS following TBI.

## Materials and methods

We performed a systematic review and NMA in accordance with the Preferred Reporting Items for Systematic Review and Network Meta-analysis checklist [11], which was developed as a guideline for reporting research outcomes. The study protocol was registered at PROSPERO (CRD42020172968).

### Search strategy and study selection

We searched PubMed, Embase, Scopus, and the Cochrane Central Register of Controlled Trials from database inception to July 13, 2021. The keyword combinations used were as follows: "traumatic brain injury" AND ("seizure" OR "post trauma seizure") AND "anticonvulsant drugs." Search strategies for each database are provided in S1 Table [12]. We included RCTs enrolling adults or children with TBI and comparing the efficacy of an AED with another AED or placebo in preventing early and late PTS. "Early PTS" refers to the occurrence of seizures within 1 week of head injury, and "late PTS" refers to seizures occurring 1 week or later after head trauma. We also searched Clinical Trials (www.clinicaltrials.gov) for ongoing trials, and reviewed reference lists from included and other related studies. No limitations in language were applied. We searched relevant articles from inception until July 13, 2021. AEDs of

different doses (flexible or fixed) were considered identical treatments. Two reviewers (B.C.W. and C.J.L.) independently screened all titles and abstracts according to the inclusion criteria. Any disagreements were resolved through discussion with a third reviewer (P.Y.C.).

## Data extraction and methodological assessment of study quality

Two authors (B.C.W and C.J.L.) independently extracted the data from each included study by using a predesigned form. The extracted data included the name(s) of the author(s), year of publication, study country, age, male percentage, sample size, drug use details (type, dosage, and frequency), AED prophylaxis, duration of follow-up, and outcomes (early and late PTS). Disagreements were resolved through consensus.

Two reviewers (B.C.W. and C.J.L) independently evaluated the risk of bias for each study by using version 2 of the Cochrane tool for assessing risk of bias in randomized trials (RoB 2.0) [13]. Five key domains of bias were evaluated: (1) bias arising from the randomization process; (2) bias due to deviations from intended interventions; (3) bias due to missing outcome data; (4) bias in measurement of the outcome; and (5) bias in selection of the reported result [13]. Responses to each domain resulted in an overall risk of bias for each study, which was judged to be low, high, or of some concern. Any discrepancies were resolved through discussion leading to consensus.

## Data analysis

All statistical analyses were performed using the STATA statistical software package (version 14). Odds ratios (ORs) with 95% confidence intervals (CIs) were calculated for early and late PTS. DerSimonian and Laird random-effects models [14] were used to perform conventional pairwise meta-analyses to directly compare any two AEDs. The Cochrane Q test, with statistical significance set at $p < 0.1$, was used to test between-study heterogeneity. Heterogeneity across head-to-head trials was assessed using the $I^2$ statistic, with values $> 50\%$ roughly indicating substantial heterogeneity [15]. Publication bias was assessed using Egger's test [16], with the statistical significance level set at 5%. The NMA was conducted under a frequentist framework with a random-effects model (using the suite of network commands written by Ian White) [17]. We assessed statistical inconsistency by using the loop-specific approach, side-splitting model, and design-by-treatment interaction model [17]. We ranked probabilities for each treatment, and the surface under the cumulative ranking (SUCRA) curve was employed to facilitate the evaluation of relative treatment efficacy [16, 18]. A higher SUCRA score (range, 0%–100%) indicated a greater likelihood of therapy in the top rank or one of the top ranks [19]. Sensitivity analysis was performed by excluding the study by Young et al. [20], which enrolled only patients younger than 10 years old.

We evaluated the proportion of direct and indirect evidence contributing to each comparison by using the direct evidence plot. According to König et al. [21], minimal parallelism and mean path length are used to estimate the degree of indirectness in the reported pooled outcome, with lower values for minimal parallelism and mean path length $> 2$ indicating that results for a specific comparison should be interpreted with caution because additional similarity assumptions must be made when each direct comparison is serially combined. Transitivity, which refers to the absence of systematic differences between the comparisons of interest other than the treatments being compared, is a crucial assumption in NMA [22]. Transitivity was qualitatively assessed by comparing the distribution of potential effect modifiers, including age and male percentage, across various comparisons in the network; if substantial variation between comparisons on effect modifiers was identified, intransitivity was suggested [22].

We used the confidence in network meta-analysis (CINeMA) web application, which is based on the Grading of Recommendations, Assessment, Development and Evaluation method, to assess the certainty of evidence produced by the synthesis for outcomes of early and late PTS. The CINeMA framework accounts for six domains that affect confidence levels in NMA results: within-study bias, reporting bias, indirectness, imprecision, heterogeneity, and incoherence. Each domain is rated as "no concerns," "some concerns"," or "major concerns," except for reporting bias, which is graded as "suspect" or "undetected." Judgments are then summarized across the domains as "high," "moderate," "low," and "very low" for each treatment comparison [23, 24]. For imprecision, the threshold was set at an OR of 1.05 for outcome comparisons after discussion.

## Results

### Study selection and characteristics

A total of 749 articles were identified and retrieved using the described search methodology, with two more articles retrieved manually after the reference lists of other relevant review articles had been reviewed. All articles were then imported into Endnote X7. Using the Endnote X7 "Find duplicate" function, we excluded 206 duplicate articles. The titles and abstracts of the remaining 543 articles were examined and then screened against the predetermined inclusion and exclusion criteria. After all ineligible articles were excluded, 10 articles were reviewed in full text. One of the 10 articles was excluded because it compared phenobarbital use with usual care, which could not connect with the network map [25]. Two additional articles were identified through a reference list search of the included studies. Eventually, 11 articles with 2,450 head injury cases were included for the analyses (Fig 1). Of these, seven articles were from the United States [20, 25–30], one was from the United Kingdom [31], one was from Germany [26], one was from France [27], and one was from Pakistan [32]. Among the included articles, five studies enrolled participants with severe TBI and four studies recruited patients with moderate to severe TBI. Most of the studies (n = 8) included more male patients. Most studies enrolled patients with a wide age range (toddlers/adolescents to older patients), except for the study of Young et al. [20], which enrolled only patients younger than 10 years old.

Among the included articles, six articles were related to the prevention of both early and late PTS [26–28, 33–35], three articles were related to the prevention of only early PTS [20, 29, 32], and the remaining two articles were related to the prevention of only late PTS [30, 31].

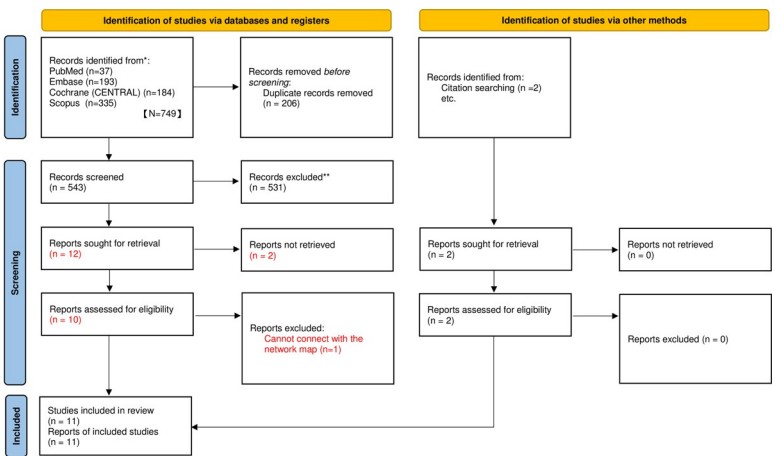

**Fig 1. Flow diagram of included studies.**

Nine articles compared phenytoin to either placebo [20, 27, 29–31, 34] or other AEDs [32, 33, 35], and one article each compared carbamazepine [26] and magnesium sulfate (MgSO$_4$) [28] to placebo (S2 Table).

The transitivity of potential effect modifiers (age and male percentage) is presented in S1 Fig. Their distributions were not significantly different across treatments, except for Young's (2004) study [20] for young age and Khan's (2016) study [32] for low male percentage, indicating potential threats to the transitivity assumption.

## Network plots

Regarding early PTS, head-to-head comparisons across nine RCTs comprising six treatments (i.e., placebo, carbamazepine phenytoin, levetiracetam, valproate, and MgSO$_4$) and 2,071 patients were performed using NMA (Fig 2A). The network plot of head-to-head comparisons

(A)

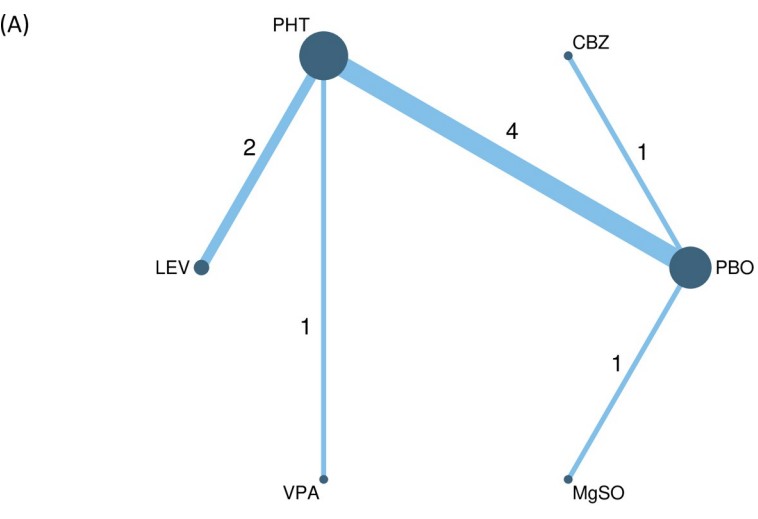

(B)

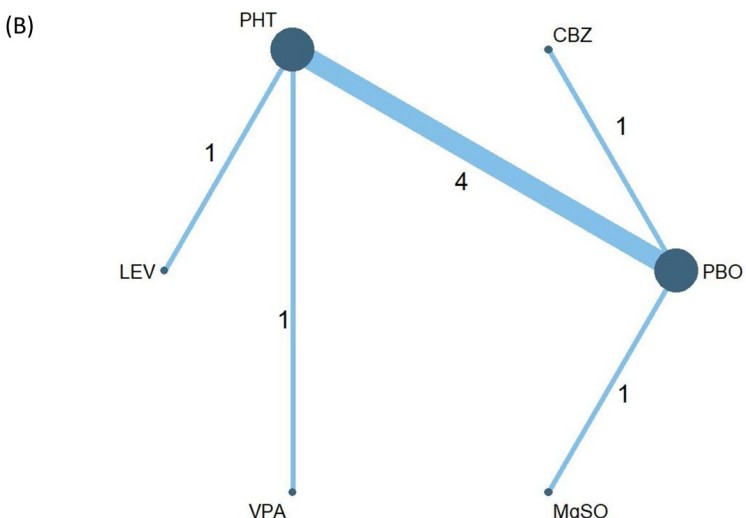

**Fig 2.** (A). Network maps for comparing different AEDs on preventing early post-traumatic seizures. (B). Network maps for comparing different AEDs on preventing late post-traumatic seizures.

of AEDs for late PTS included eight RCTs comprising six treatments (i.e., placebo, carbamaze-pine, phenytoin, levetiracetam, valproate, and $MgSO_4$) and 1,788 patients (Fig 2B). In the two network maps, the comparison between placebo and PTH was the most compared treatment arm. Because the two network plots did not form a closed loop, inconsistency was not evaluated.

### Effects of AEDs on early and late PTS

S3 Table presents the results of the pairwise meta-analysis and heterogeneity estimates. In terms of early PTS, only two pairwise interventions compared with placebo groups had two or more studies: phenytoin vs. placebo and levetiracetam vs. phenytoin. The remaining pairwise interventions included single studies. No significant difference was observed between treatment comparisons, except for carbamazepine vs. placebo (OR = 0.29, 95% CI = 0.12–0.71, n = 1), indicating a greater reduction in attenuated early PTS. For late PTS, only the pairwise comparison between phenytoin and placebo had four studies. No AEDs had any significant treatment effects. Next, we compared the combined available AEDs with placebo for early and late PTS; the pooled ORs were 0.42 and 0.82, respectively, estimated using a random-effects model (95% CI = 0.21–0.82 and 0.47–1.43, S3 Table). This indicates that AEDs as a whole significantly reduced early PTS but not late PTS compared with placebo.

Regarding the results of NMA on early PTS (S4 Table), carbamazepine (OR = 0.29, 95% CI = 0.07–1.18), phenytoin (OR = 0.43, 95% CI = 0.18–1.01), and levetiracetam (OR = 0.56, 95% CI = 0.12–2.55) were more likely to reduce early PTS compared with placebo; however, the effects were not significant (S4 Table). Furthermore, no AED was superior to another. Heterogeneity was assessed using the $I^2$ test (29.5%, $p$ = 0.16) and tau-squared test (0.30). The SUCRA analysis (S2 Fig) indicated that carbamazepine ranked first (82.3%), followed by phenytoin (74.4%) and levetiracetam (57.2%). The Egger's test for studies with small sample sizes returned a $p$-value of 0.457, indicating no publication bias.

Compared with the placebo, carbamazepine (OR = 0.64, 95% CI = 0.08–5.28) and phenytoin (OR = 0.71, 95% CI = 0.23–2.20) were more likely to reduce the risk of late PTS, but the treatment effects were nonsignificant (S4 Table). No one AED was superior to any other. Heterogeneity between studies was observed ($I^2$ = 72.7%, $p$ = 0.02; tau squared = 0.60). The SUCRA analysis (S2 Fig) indicated that phenytoin ranked the highest (62.7%), followed by carbamazepine (60.0%) and $MgSO_4$ (48.0%). The Egger's test for studies with small sample sizes returned a $p$-value of 0.24, indicating no publication bias.

Of the 15 unique treatment comparisons, only 5 contained entire direct evidence (proportion = 100%), whereas the remaining 10 were based entirely on indirect evidence (S3 Fig). For all estimates, the minimum number of independent paths contributing to the effect size estimate on an aggregated level (minimal parallelism) was 1, implying less robust estimates. We observed mean path lengths >2 in 4 of the 15 estimates; this implies that the estimate in question should be interpreted with caution.

### Sensitivity analysis

After the study of Young et al., which involved only patients younger than 10 years, was excluded [20], phenytoin significantly reduced the occurrence of early PTS compared with placebo (OR = 0.33; 95% CI = 0.14–0.78; S4 Table).

### Adverse events

Three of the included studies reported the occurrence of any adverse events [31, 33, 34] and two of them reported skin rash after the use of AEDs [31, 34]. The occurrence of any adverse

events was 13.9% with phenytoin, 9.42% with placebo, and 6.48% with valproate, and that of skin rash was 10.3% with phenytoin and 6.5% with placebo. Mortality was the most frequently documented adverse event (8 of 10 studies). Levetiracetam was associated with the highest mortality rate (41.18%), followed by carbamazepine (36.0%), $MgSO_4$ (18.03%), placebo (16.69%), phenytoin (13.94%), and valproate (12.96%). Because of the limited number of studies reporting any adverse events and skin rash, comparative NMA was performed only for mortality. As presented in S4 Fig, the higher mortality rate was comparable between AEDs.

## Risk of bias assessment and confidence in evidence

We identified four studies with a high risk of overall bias [26, 27, 29, 30]. Regarding the randomization process of the included studies, one study was graded as being at high risk of bias because its allocation was based on participant admission on either odd or even days [27], and four were graded as being at unknown risk because their methods of allocation concealment were not clearly described [26, 29, 30, 34]. Regarding bias due to deviation from intended interventions, one study was rated as high risk because research personnel may have been aware of the group assignment [27] (S5 Table).

The grading of the comparisons with CINeMA revealed mainly low to very low confidence ratings. This was due to concerns about within-study bias due to poor reporting of the randomization and blinding procedures. Evidence of imprecision was noted, likely because of the low number of trials available for comparison. Owing to the star-shaped network, the evidence quality in the domain of incoherence bias was downgraded (S6 Table).

## Discussion

In this NMA, we included 11 studies enrolling 2,450 patients, which compared various prophylactic AEDs for preventing early and late PTS in patients with TBI. Our NMA findings indicated that no AED was superior to any other in preventing early or late PTS. However, our sensitivity analysis revealed that phenytoin might reduce early PTS compared with placebo, which is consistent with the Brain Trauma Foundation *Guidelines for the Management of Severe Traumatic Brain Injury (Fourth Edition)* [7]. Notably, the results of our pairwise meta-analysis are partially similar to those of a pairwise meta-analysis by Thompson et al. (2015), who compared the efficacy of traditional AEDs in preventing early PTS following head injury with that of a placebo. Their results indicated that AEDs, namely carbamazepine, phenobarbital, phenytoin, levetiracetam, and valproate, effectively reduced the occurrence of early PTS following head injury compared with placebo. However, no statistically significant differences were observed when phenytoin and other AEDs, including levetiracetam and valproate, were compared [9]. The results of a meta-analysis by Zafar and colleagues in 2012 demonstrated no statistically significant differences between the prophylactic use of phenytoin and that of levetiracetam in reducing the occurrence of early PTS following head injury, further supporting our study results [8]. The inconsistency in the results of the two studies may be because the AEDs were pooled together using pairwise meta-analyses in both studies. The two pairwise meta-analyses [8, 9] pooled relevant studies together, thereby substantially increasing the statistical power. By contrast, NMA, which our study adopted, may have led to few treatment comparison studies, causing reduced statistical power [36]. Our study results should therefore be interpreted with caution, with future studies required to verify our findings.

According to the results of the sensitivity analysis and transitivity assessment, age potentially affects the treatment effect of anticonvulsant drugs on PTS. We observed that after the study of Young et al. [20], phenytoin exhibited higher efficacy in preventing early PTS. The influence of age on the development of PTS has been debated, with some studies suggesting

that young children (<2 years old) are more likely to develop PTS than older children [37–39] and some rejecting the association [40]. In fact, pediatric PTS is still not well understood, and studies on the effects of AEDs on pediatric PTS are still scarce. Further studies are warranted to explore how age affects the efficacy of AEDs in patients with PTS.

Similar to the results of previous conventional meta-analyses [8, 9], this study revealed that treatment with each AED compared to other AEDs or placebo could not effectively reduce the occurrence of late PTS following head injury. In clinical practice, patients often stop taking drugs or forget to take drugs because of remission of their symptoms, which leads to insufficient drug concentrations in their blood [41]. However, most studies included in our analysis did not clearly describe how patients were followed for medication use after their discharge from the hospital, including confirming patients' medication compliance, monitoring the drug concentration in blood, and adjusting drug dosages. As a result, the AEDs could have failed to effectively prevent the occurrence of late PTS following head injury.

Several limitations should be considered when interpreting our data. First, the study included patients with a wide age range (from children to older patients), which may have reduced the internal validity of the study. Second, only a few studies were included for single study arm comparison in the NMA, thus limiting the statistical power. Third, the CINeMA results and direct evidence proportion plots suggested that our evidence has low or very low quality and less reliable estimates; therefore, our data should be interpreted with caution. Finally, not all included studies provided clear details of medication administration, including loading dose, maintenance dose, and monitoring and target ranges of drug concentrations in blood. As a result, the therapeutically effective dose of the drugs could not be correctly estimated in the included studies, which could have caused either persistent epilepsy due to poor efficacy resulting from insufficient drug concentrations in blood or adverse reactions due to drug overdose.

## Conclusion

Our pairwise MA suggests that AEDs as a whole are superior to placebo in preventing early PTS; however, there is currently no strong evidence to support any specific medication in our NMA. Phenytoin has the largest body of evidence for its efficacy and performed well in this meta-analysis despite the lack of statistical significance. Our results support the treatment guidelines regarding the prophylactic use of phenytoin in preventing early PTS, but the optimal medication is unclear. PTS is a possible complication that affects patients with head injuries. Clinicians should be alerted to our findings, and future high-quality RCTs are warranted to examine the effects of AEDs on early and late PTS, especially in children.

## Supporting information

**S1 Fig. The transitivity analysis of potential effect modifiers.** (A) age and (B) male percentage. 1 indicates Glotzner et al., (1983) study; 2 indicates Pechadre et al., (1991) study; 3 indicates Szaflarski et al., (2010) study; 4 indicates Temkin et al., (1990) study; 4 indicates Temkin et al., (1999) study; 6 indicates Temkin et al., (2007) study; 7 indicates Young et al., (1983) study; 8 indicates Young et al., 2004) study; 9 indicates Khan et al., (2016) study. CBZ = carbamazepine, LEV = levetiracetam, VPA = valproate, PBO = placebo, PTH = phenytoin, MgSO = Magnesium Sulfate (MgSO4).
(TIF)

**S2 Fig. The surface under the cumulative ranking of anticonvulsant drugs according to early and late post-traumatic seizure.** The surface under the cumulative ranking curve value is the probability each intervention has of being among the best in the network, with larger

values representing higher ranking probabilities.
(TIF)

**S3 Fig.** Direct evidence plots for (A) early post-traumatic seizure and (B) late post-traumatic seizure. This figure shows the proportion of direct evidence (orange shade) in overall reporting of network value (blue shade). Minimal parallelism: Bar chart displaying the minimum number of independent paths contributing to the effect estimate on an aggregated level. All interventions included in our analysis have values equal to one, highlighting that estimates had contributions that extended beyond single pair-wise comparisons. Mean path length. Bar chart displaying the mean path length, which characterizes the degree of indirectness of an estimate. Higher mean path lengths indicate less reliable estimates. Mean path length for indirect comparison was equal to or above 2, which points to a less reliable estimate.
(TIF)

**S4 Fig. The forest plots of AEDs comparisons for mortality.**
(TIF)

**S1 Table. An example of search strategies.**
(DOCX)

**S2 Table. Characteristics of studies.**
(DOCX)

**S3 Table. Pairwise meta-analytic results for early and late posttraumatic seizures.**
(DOCX)

**S4 Table. League tables for comparing anticonvulsant drugs for early and late post-traumatic seizures.**
(DOCX)

**S5 Table. Risk of methodological bias score of the studies.**
(DOCX)

**S6 Table. CINeMA summary tables.**
(DOCX)

## Author Contributions

**Conceptualization:** Bo-Cyuan Wang, Chia-Jou Lin.

**Data curation:** Hui-Tzung Luh, Shu-Hua Hsieh, Ting-Jhen Chen, Chia-Rung Wu.

**Formal analysis:** Bo-Cyuan Wang, Hsiao-Yean Chiu.

**Methodology:** Bo-Cyuan Wang, Hsiao-Yean Chiu.

**Software:** Hsiao-Yean Chiu.

**Supervision:** Pin-Yuan Chen.

**Writing – original draft:** Bo-Cyuan Wang, Hsiao-Yean Chiu.

**Writing – review & editing:** Bo-Cyuan Wang, Hsiao-Yean Chiu, Pin-Yuan Chen.

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
