## [Decision Letter · Decision Letter 0]

4 May 2021

PONE-D-21-09280

Efficacy and safety of prophylactic anticonvulsant drug following traumatic brain injury: a systematic review and network meta-analysis of randomized controlled trials

PLOS ONE

Dear Dr. Chen,

Thank you for submitting your manuscript to PLOS ONE. After careful consideration, we feel that it has merit but does not fully meet PLOS ONE’s publication criteria as it currently stands. Therefore, we invite you to submit a revised version of the manuscript that addresses the points raised during the review process.

We look forward to receiving your revised manuscript.

Kind regards,

Giuseppe Biagini, MD

Academic Editor

PLOS ONE

Journal Requirements:

2. Please include your tables as part of your main manuscript and remove the individual files. Please note that supplementary tables should remain uploaded as separate "supporting information" files.

Reviewers' comments:

Reviewer's Responses to Questions

**Comments to the Author**

1. Is the manuscript technically sound, and do the data support the conclusions?

Reviewer #1: Partly

Reviewer #2: Yes

2. Has the statistical analysis been performed appropriately and rigorously? 

Reviewer #1: I Don't Know

Reviewer #2: Yes

3. Have the authors made all data underlying the findings in their manuscript fully available?

Reviewer #1: Yes

Reviewer #2: Yes

4. Is the manuscript presented in an intelligible fashion and written in standard English?

Reviewer #1: No

Reviewer #2: Yes

5. Review Comments to the Author

Reviewer #1: This manuscript is a network meta-analysis (NMA) of randomised controlled trials performed to evaluate the effect of prophylactic anticonvulsant therapy in patients with traumatic brain injury. The topic is of interest and certainly an area of equipoise in current practice. Current guidelines suggest the use of empiric anticonvulsant therapy in patients presenting with severe traumatic brain injury for up to one week, but practice has been demonstrated in surveys to vary widely among neurosurgeons. The authors’ analysis of the twelve included studies suggests no benefit to any individual antiepileptic drug.

Overall, this is a useful and important topic with interesting implications that might be somewhat answered using an NMA. However, there are methodological flaws which should be addressed before publication can be considered.

Statistical analysis: Lacking in detail, see comments below. However, the treatment ranking described by the authors is reproducible from the data provided.

English language/grammar: Requires further proof-reading.

Major comments:

1. Firstly, the authors should consider whether an NMA is appropriate. I note that, of the 5 drugs studied for the prevention of early seizures for example, 4 were compared in only a single study arm which limits power somewhat. An NMA, if performed, should be very carefully evaluated for integrity in such an instance.

2. How were early and late seizures defined? This is important for interpretation and also ensuring transitivity in the NMA and as such should be clearly stated in the methods section.

3. Transitivity is perhaps the most crucial assumption of the NMA and should be defined and assessed by the authors. This should be explained in the methods section.

4. The exact search strategy used should be provided (as supplementary) for each database in such a way that it is fully reproducible. This should ideally comply with the PRISMA-P statement.

5. The authors report to have assessed risk of bias in individual studies using the Cochrane Handbook, providing a reference from 2011. Risk of bias in randomised trials is more appropriately assessed using an updated, validated tool such as the RoB 2.0.

6. The authors report to have performed pairwise meta-analysis in addition to NMA for each pairing, but do not appear to have reported the results. Pairwise analyses should be reported separately from and clearly delineated from the network estimates derived from the NMA, perhaps in league table form.

7. The authors mention that random effects regression was applied to the pairwise analyses. Was a random effects model also used for the network estimates? This should be clarified.

8. The authors mention the calculation of a SUCRA treatment ranking and SUCRA ranking positions are alluded to in the manuscript but the treatment ranking for each category studied is not overtly provided. These rankings should be included in tabular form for each analysis with the exact SUCRA scores provided. A cumulative ranking plot may also be of benefit to illustrate.

9. SUCRA is a useful metric in the NMA but NMA is a relatively novel concept with which many readers will be unfamiliar and its calculation, implications and interpretations should therefore be explained in the methods.

10. On page 8, the authors mention that Egger’s p < 0.05 indicated no significant publication bias. This is incorrect - p < 0.05 indicates significant asymmetry and suggests the presence of publication bias.

11. The full results of the NMA including effect size estimates with associated confidence intervals and SUCRA scores should be reported for each analysis separately in the text and in the abstract.

12. The study by Manaka (1992) refers to a control group consisting of “usual care”, but the authors mention that some patients received anticonvulsants. There is no node relating to “usual care” in the network plots – was this group considered equivalent to placebo control? If so, this should be overtly stated as this may violate the transitivity assumption of the NMA.

13. The authors mention performing meta-regression in the methods (page 6) but no further detail was provided. If this was performed, please provide details of the statistical methods and software used for this in the methods along with results including regression coefficients, residual heterogeneity and significance levels in the results section.

14. The exploration of inconsistency is mentioned. Where available, the results of the node-splitting analysis performed should be provided for each comparison in the network. Heterogeneity within designs and between designs should be clearly quantitatively reported for each analysis.

15. The forest plots are currently rather illegible. The axes should be decompressed as the numbers are currently unreadable and the treatment identifiers (A, B, etc) should probably just be substituted for the full treatment names or three-letter identifiers for each treatment eg. PHT for phenytoin. In addition, the legend is absent from figure 5.

16. The authors conclude by stating that clinicians should incorporating their findings, that no anticonvulsant drug significantly decreases the likelihood of experiencing post-traumatic epilepsy, into clinical practice. I find this excessively assured. The authors did not attempt to pool all anticonvulsants into a pairwise anticonvulsants versus placebo meta-analysis, and a Cochrane review on the topic has demonstrated a lower likelihood of seizures when all anticonvulsants are considered together. It is possible, and even likely, that studies of individual treatments are underpowered to demonstrate an effect but when pooled demonstrate that prophylactic anticonvulsant therapy may in fact decrease the likelihood of post-traumatic seizures. The authors refer to this on page 9-10, stating that the study has better understanding of the individual treatments as a result, but consideration should also be given to the increased statistical power achieved by pooling the individual therapies.

Minor comments:

1. The authors should be commended for reporting raw data in Table 1, but this should be reported separately also as supplementary, ideally in spreadsheet format to allow easier reproduction of the results.

2. I find the choice of skin rash as the only adverse effect to be investigated unconvincing. While it may be the most common, its clinical relevance in patients with severe traumatic brain injury is questionable and perhaps more serious adverse effects should also be considered despite being less common? If this is retained, it should be defined – skin rash associated with anticonvulsants ranges from mild cutaneous irritation to life-threatening dermatological syndromes.

3. There are some references missing in the discussion wherein authors are mentioned but no superscript reference is provided eg. by Zafar et al. on page 9.

4. The authors should consider evaluating mean path length(1) given that many treatments are represented by a single study arm.

References:

1. König J, Krahn U, Binder H. Visualizing the flow of evidence in network meta-analysis and characterizing mixed treatment comparisons. Stat Med. 2013 Dec 30;32(30):5414-29. doi: 10.1002/sim.6001. Epub 2013 Oct 4. PMID: 24123165.

Reviewer #2: In this interesting systematic review the authors aimed was to systematically compare the effects and safety of prophylactic anticonvulsant drug use in patient with traumatic brain injury. Network meta-analysis using multivariate random effects and frequentist framework was applied. A total of 12 randomized controlled trials involving 1,431 participants were included. The authors underline that this study suggests no beneficial effect of anticonvulsant drug on early and late post-traumatic seizure, which does not support the recommendations of the clinical guideline of using Phenytoin as the first-line therapy in treatment of early post-traumatic seizure.

The manuscript is well written and it could have a relevant impact on the readership. However, there are few minor points that need to be clarified.

Specific comments:

- The sentence reported in the abstract “In comparison with placebo, phenytoin, valproate, levetiracetam, carbamazepine, and MgSO4 could significantly reduce the early post-traumatic seizure; and phenytoin, valproate, levetiracetam, carbamazepine, MgSO4, and phenobarbital were not significantly superior to placebo.” is not clear; please reformulate it.

- page 9: please change “seizure” with “seizures”

- Please check the different values included in the flow diagram of the study selection process.

6. PLOS authors have the option to publish the peer review history of their article (what does this mean?). If published, this will include your full peer review and any attached files.

Reviewer #1: **Yes: **Jack Henry

Reviewer #2: No

---

## [Author Response · Author response to Decision Letter 0]

16 Sep 2021

Reviewer #1: This manuscript is a network meta-analysis (NMA) of randomised controlled trials performed to evaluate the effect of prophylactic anticonvulsant therapy in patients with traumatic brain injury. The topic is of interest and certainly an area of equipoise in current practice. Current guidelines suggest the use of empiric anticonvulsant therapy in patients presenting with severe traumatic brain injury for up to one week, but practice has been demonstrated in surveys to vary widely among neurosurgeons. The authors’ analysis of the twelve included studies suggests no benefit to any individual antiepileptic drug. Overall, this is a useful and important topic with interesting implications that might be somewhat answered using an NMA. However, there are methodological flaws which should be addressed before publication can be considered.

Response: Thank you for your thoughtful review. We have carefully studied your comments and suggestions and revised our paper accordingly. The following are our point-by-point responses to your specific comments. We hope our revisions are acceptable and our responses adequately address the comments. Thank you for your consideration.

Statistical analysis: Lacking in detail, see comments below. However, the treatment ranking described by the authors is reproducible from the data provided.

Response: Thank you for the comment. We have revised the statistical analysis according to your suggestion.

English language/grammar: Requires further proof-reading.

Response: Thank you for the comment. We have had the manuscript reviewed by a native English speaker with scientific expertise to ensure proper English grammar and usage. 

Major comments:

1. Firstly, the authors should consider whether an NMA is appropriate. I note that, of the 5 drugs studied for the prevention of early seizures for example, 4 were compared in only a single study arm which limits power somewhat. An NMA, if performed, should be very carefully evaluated for integrity in such an instance.

Response: Thank you for the thoughtful comment. We agree with your concerns that the NMA may have caused reduced statistical power, thus threatening the internal validity of our findings. However, compared with traditional pairwise meta-analysis, the strength of the NMA lies in comparing different AEDs simultaneously. Even so, we have emphasized this concern in the section of Discussion and as one of the study limitations as follows: It is worth noting that the two pairwise meta-analyses [8, 9] that pooled relevant studies together may have substantially increased statistical power by doing so. By contrast, NMA, which our study adopted, may have led to few treatment comparison studies, causing reduced statistical power [33]. Our study results should therefore be interpreted with caution, with future studies required to verify our findings (Discussion section); Several limitations should be considered when interpreting our data. First,…Second, only a few studies were included for single study arm comparison, thus limiting the statistical power.…(Limitations)

2. How were early and late seizures defined? This is important for interpretation and also ensuring transitivity in the NMA and as such should be clearly stated in the methods section.

Response: We agree with your comment that clear definitions of early and late seizures are critical for interpretation and ensuring transitivity in the NMA. We have added the definition in the Material and Methods section as follows (P.4): “Early PTS” refers to the occurrence of seizures within 1 week of head injury, and “late PTS” refers to seizures occurring 1 week or later after head trauma.

3. Transitivity is perhaps the most crucial assumption of the NMA and should be defined and assessed by the authors. This should be explained in the methods section.

Response: We have defined and assessed transitivity, the most crucial assumption of the NMA in the methods section as follow (p 7): Transitivity, which refers to an absence of systematic differences between the available comparisons other than the treatments being compared, is a crucial assumption for an NMA [20]. Transitivity was qualitatively assessed by the authors; if a substantial variation between comparisons on effect modifiers was identified, intransitivity was suggested [20]. We have also qualitatively described the details of the included studies to confirm the issue of transitivity. We observed that age variations may cause intransitivity; hence, sensitivity analysis was further performed. All details can be found in the Results section (page 8 to 9).

4. The exact search strategy used should be provided (as supplementary) for each database in such a way that it is fully reproducible. This should ideally comply with the PRISMA-P statement.

Response: Thank you for the comments. We have provided the exact search strategy (as Supplementary table 1) for each database, which ideally complies with the PRISMA-P statement.

5. The authors report to have assessed risk of bias in individual studies using the Cochrane Handbook, providing a reference from 2011. Risk of bias in randomised trials is more appropriately assessed using an updated, validated tool such as the RoB 2.0.

Response: Thank you for the comments. We have reassessed the study quality using RoB 2.0. All changes are marked in red in both the Material and Methods and the Results sections.

6. The authors report to have performed pairwise meta-analysis in addition to NMA for each pairing, but do not appear to have reported the results. Pairwise analyses should be reported separately from and clearly delineated from the network estimates derived from the NMA, perhaps in league table form.

Response: Thank you for the comments. We did perform the pairwise meta-analysis before the NMA. We have added the pairwise analyses in a league table format.

7. The authors mention that random effects regression was applied to the pairwise analyses. Was a random effects model also used for the network estimates? This should be clarified.

Response: Thank you for the comments. This was a typo. We have deleted the sentence describing random effects regression and noted the random-effects model in the Data analysis section as follows: The network meta-analysis was conducted under a frequentist framework with a random-effects model. (Page 7)

8. The authors mention the calculation of a SUCRA treatment ranking and SUCRA ranking positions are alluded to in the manuscript but the treatment ranking for each category studied is not overtly provided. These rankings should be included in tabular form for each analysis with the exact SUCRA scores provided. A cumulative ranking plot may also be of benefit to illustrate.

Response: Thank you for your suggestions. We have added SUCRA plots with exact SCURA scores for each outcome in Supplementary Figure S1.

9. SUCRA is a useful metric in the NMA but NMA is a relatively novel concept with which many readers will be unfamiliar and its calculation, implications and interpretations should therefore be explained in the methods.

Response: Thank you for your suggestions. We have added the implications and interpretations of the SUCRA score in the section of Data Analysis as follows: We ranked probabilities for each treatment, and the surface under the cumulative ranking (SUCRA) curve was employed to facilitate the evaluation of relative treatment efficacy [16, 18]. A higher SUCRA score (range, 0%–100%) indicated a greater likelihood of therapy in the top rank or one of the top ranks [19].

10. On page 8, the authors mention that Egger’s p < 0.05 indicated no significant publication bias. This is incorrect - p < 0.05 indicates significant asymmetry and suggests the presence of publication bias.

Response: It was a typo. We have provided the exact p-value indicative of no significant publication bias.

11. The full results of the NMA including effect size estimates with associated confidence intervals and SUCRA scores should be reported for each analysis separately in the text and in the abstract.

Response: Thank you for your suggestion. We have added the effect size estimates with associated CIs and SUCRA scores for each outcome separately in the main text and abstract.

12. The study by Manaka (1992) refers to a control group consisting of “usual care”, but the authors mention that some patients received anticonvulsants. There is no node relating to “usual care” in the network plots – was this group considered equivalent to placebo control? If so, this should be overtly stated as this may violate the transitivity assumption of the NMA.

Response: Thank you for your suggestion. After reviewing the article, we found that usual care arm is not equivalent to placebo, causing the study (Manaka [1992]) could bot connect with NMA map. We therefore removed it from final analysis.

13. The authors mention performing meta-regression in the methods (page 6) but no further detail was provided. If this was performed, please provide details of the statistical methods and software used for this in the methods along with results including regression coefficients, residual heterogeneity and significance levels in the results section.

Response: Thank you for your suggestions. We have deleted this sentence because we did not perform meta-regression due to the inclusion of small-size number of studies.

14. The exploration of inconsistency is mentioned. Where available, the results of the node-splitting analysis performed should be provided for each comparison in the network. Heterogeneity within designs and between designs should be clearly quantitatively reported for each analysis.

Response: Thank you for your suggestions. Because the network plots did not form a closed loop, the pairwise comparisons were based on direct or indirect evidence. That is, inconsistency was not evaluated for the networks. We have mentioned this point in the Network Plots section.

15. The forest plots are currently rather illegible. The axes should be decompressed as the numbers are currently unreadable and the treatment identifiers (A, B, etc) should probably just be substituted for the full treatment names or three-letter identifiers for each treatment eg. PHT for phenytoin. In addition, the legend is absent from figure 5.

Response: Thank you for the comments. To enhance the readability of our data, we have replaced the forest plots with league tables to present the results of treatment comparisons.

16. The authors conclude by stating that clinicians should incorporating their findings, that no anticonvulsant drug significantly decreases the likelihood of experiencing post-traumatic epilepsy, into clinical practice. I find this excessively assured. The authors did not attempt to pool all anticonvulsants into a pairwise anticonvulsants versus placebo meta-analysis, and a Cochrane review on the topic has demonstrated a lower likelihood of seizures when all anticonvulsants are considered together. It is possible, and even likely, that studies of individual treatments are underpowered to demonstrate an effect but when pooled demonstrate that prophylactic anticonvulsant therapy may in fact decrease the likelihood of post-traumatic seizures. The authors refer to this on page 9-10, stating that the study has better understanding of the individual treatments as a result, but consideration should also be given to the increased statistical power achieved by pooling the individual therapies.

Response: We agree with your comment; hence, we have attributed increased statistical power to the pairwise meta-analyses that pooled the individual therapies, and stated that our network meta-analysis involved only a few studies for each comparison in the Discussion section as follows: It is worth noting that the two pairwise meta-analyses [8, 9] that pooled relevant studies may have substantially increased statistical power by doing so. By contrast, NMA, which our study adopted, may have led to few treatment comparison studies, causing reduced statistical power [33]. Our study results therefore should be interpreted with caution, with future studies required to verify our findings.

Minor comments:

1. The authors should be commended for reporting raw data in Table 1, but this should be reported separately also as supplementary, ideally in spreadsheet format to allow easier reproduction of the results.

Response: Thank you for the suggestion. We have provided the raw data in spreadsheet format as a supplementary Table S3.

2. I find the choice of skin rash as the only adverse effect to be investigated unconvincing. While it may be the most common, its clinical relevance in patients with severe traumatic brain injury is questionable and perhaps more serious adverse effects should also be considered despite being less common? If this is retained, it should be defined – skin rash associated with anticonvulsants ranges from mild cutaneous irritation to life-threatening dermatological syndromes.

Response: Thank you for the suggestion. After rechecking the included studies, original authors did not define skin rash in the papers; therefore, we could not identify the effects of skin rash on TBI survivors. We have deleted all relevant context regarding the adverse effects of AEDs on skin rash from the manuscript.

3. There are some references missing in the discussion wherein authors are mentioned but no superscript reference is provided eg. by Zafar et al. on page 9.

Response: We have added citation numbers for the missing references.

4. The authors should consider evaluating mean path length (1) given that many treatments are represented by a single study arm.

Response: We have added the average path length to depict the numbers of treatments in a single study arm (Figure 2).

 

Reviewer #2: In this interesting systematic review the authors aimed was to systematically compare the effects and safety of prophylactic anticonvulsant drug use in patient with traumatic brain injury. Network meta-analysis using multivariate random effects and frequentist framework was applied. A total of 12 randomized controlled trials involving 1,431 participants were included. The authors underline that this study suggests no beneficial effect of anticonvulsant drug on early and late post-traumatic seizure, which does not support the recommendations of the clinical guideline of using Phenytoin as the first-line therapy in treatment of early post-traumatic seizure. The manuscript is well written and it could have a relevant impact on the readership. However, there are few minor points that need to be clarified.

Response: Thank you for your encouraging words. We have revised our manuscript according to your suggestions, and the following are our point-by-point responses to your specific comments. We hope that our revisions are acceptable and that our responses adequately address your comments. Thank you for your consideration.

Specific comments:

- The sentence reported in the abstract “In comparison with placebo, phenytoin, valproate, levetiracetam, carbamazepine, and MgSO4 could significantly reduce the early post-traumatic seizure; and phenytoin, valproate, levetiracetam, carbamazepine, MgSO4, and phenobarbital were not significantly superior to placebo.” is not clear; please reformulate it.

Response: We have revised the abstract as follows: We found that phenytoin (odd ratios [ORs] = 0.43 and 0.71; 95% confidence intervals [CIs] = 0.18–1.01 and 0.23–2.20), levetiracetam (ORs = 0.56 and 1.58; 95% CIs = 0.12–2.55 and 0.03–84.42), and carbamazepine (OR= 0.29 and 0.64; 95% CIs = 0.07–1.18 and 0.08–5.28) were more likely to reduce early and late PTS compared with placebo; however, the treatment effects were not significant.

- page 9: please change “seizure” with “seizures”

Response: Because the abbreviation of posttraumatic seizures (i.e., PTS) was defined upon first usage, we have replaced “posttraumatic seizures” with “PTS” throughout the manuscript. 

- Please check the different values included in the flow diagram of the study selection process.

Response: We have rechecked and updated the values included in the flow diagram. All changes are marked in red.

---

## [Decision Letter · Decision Letter 1]

8 Oct 2021

PONE-D-21-09280R1Comparative efficacy of prophylactic anticonvulsant drugs following traumatic brain injury: a systematic review and network meta-analysis of randomized controlled trialsPLOS ONE

Dear Dr. Chen,

Thank you for submitting your manuscript to PLOS ONE. After careful consideration, we feel that it has merit but does not fully meet PLOS ONE’s publication criteria as it currently stands. Therefore, we invite you to submit a revised version of the manuscript that addresses the points raised during the review process.

We look forward to receiving your revised manuscript.

Kind regards,

Giuseppe Biagini, MD

Academic Editor

PLOS ONE

Reviewers' comments:

Reviewer's Responses to Questions

**Comments to the Author**

1. If the authors have adequately addressed your comments raised in a previous round of review and you feel that this manuscript is now acceptable for publication, you may indicate that here to bypass the “Comments to the Author” section, enter your conflict of interest statement in the “Confidential to Editor” section, and submit your "Accept" recommendation.

Reviewer #1: (No Response)

Reviewer #2: All comments have been addressed

2. Is the manuscript technically sound, and do the data support the conclusions?

Reviewer #1: Partly

Reviewer #2: Yes

3. Has the statistical analysis been performed appropriately and rigorously? 

Reviewer #1: No

Reviewer #2: Yes

4. Have the authors made all data underlying the findings in their manuscript fully available?

Reviewer #1: Yes

Reviewer #2: Yes

5. Is the manuscript presented in an intelligible fashion and written in standard English?

Reviewer #1: Yes

Reviewer #2: Yes

6. Review Comments to the Author

Reviewer #1: The authors should be commended for a substantial review, which has significantly improved the manuscript. The plots are significantly improved and more legible, and the writing has significantly improved. I have the following comments:

Major comments:

1. The authors now provide a definition of transitivity as suggested. However, they mention that transitivity was assessed qualitatively only (methods, page 7-8). They do not appear to have defined specific potential effect modifiers, or collected arm-level data on these modifiers. This insufficient to assess the plausibility of transitivity, especially in the context of a star-shaped network, which limits the assessment of its statistical manifestation (inconsistency). Ideally, the authors should define potential effect modifiers and compare their distribution across treatment nodes in the network.

2. On this note, the network heterogeneity, quantified by tau^2, should also be reported. I am aware that, for the late seizure analysis in particular, this will be a skewed estimate given that most treatments are comprised of only a single study. However, some effort to quantify the heterogeneity is needed.

3. The authors now report to have included the odds ratios, including direct pairwise estimates, in league table format as Table 2. This does not appear to have been provided.

4. The authors have now removed the skin rash adverse event from the NMA assessment, which seems appropriate. However, there is now no assessment of adverse events or mortality, which is a significant limitation for any clinician reading, because this informs treatment choice as much as, or even more than efficacy. Ideally, the authors should record data on the number of adverse events recorded and the mortality rate between the various treatments evaluated. This may not be appropriate for network meta-analysis, but should at least be recorded, displayed and discussed.

5. The authors claim to now have provided the mean path length for comparisons. What is provided in the plots is in fact the number of studies forming the comparison, as far as I can see. The mean path length I refer to is as described by Konig et al. (2013), below.

6. It is not emphasised enough that anticonvulsants as a whole are superior to placebo for the prevention of early seizures. This has been strongly shown by the Cochrane review. In addition, if studies comparing an antiepileptic drug to placebo in the authors' provided data are meta-analysed (any AED vs. placebo) the summary result from a random effects model is OR 0.42, 95%CI 0.42, p = 0.01, I^2 = 37.5%, tau^2 = 0.25. To me, this is a central point that should be better emphasised by the authors. When this meta-analysis is also performed, the findings suggest that anticonvulsants as a whole are superior to placebo, but the various drugs appear to have equivalent efficacy.

7. In my opinion, the conclusion that PHT may be a beneficial treatment, without reference to the aggregate of anticonvulsant drugs, is rather unsupported as a result. This is especially given the case that phenytoin did not rank highest in the NMA, did not have superior efficacy to any of the individual drugs in the point estimates and was only superior to placebo in a seemingly arbitrary sensitivity analysis.

8. In line with recent updates to the PRISMA guidelines, the authors confidence in their findings should be discussed. In network meta-analysis, the CINeMA tool provides a framework and associated web application for this (referenced below).

König J, Krahn U, Binder H. Visualizing the flow of evidence in network meta-analysis and characterizing mixed treatment comparisons. Stat Med. 2013 Dec 30;32(30):5414-29. doi: 10.1002/sim.6001. Epub 2013 Oct 4. PMID: 24123165.

Nikolakopoulou A, Higgins JPT, Papakonstantinou T, Chaimani A, Del Giovane C, Egger M, Salanti G. CINeMA: An approach for assessing confidence in the results of a network meta-analysis. PLoS Med. 2020 Apr 3;17(4):e1003082. doi: 10.1371/journal.pmed.1003082. PMID: 32243458; PMCID: PMC7122720.

Reviewer #2: The Authors have edited the manuscript as suggested, furthermore I have no further comments to add .

7. PLOS authors have the option to publish the peer review history of their article (what does this mean?). If published, this will include your full peer review and any attached files.

Reviewer #1: **Yes: **Jack Henry

Reviewer #2: No

---

## [Author Response · Author response to Decision Letter 1]

10 Feb 2022

Thank you for your valuable comments that have helped improve our manuscript. We have carefully studied your comments and suggestions and revised our paper accordingly. The following are our point-by-point responses to your specific comments. We hope that our revisions are acceptable and that our responses adequately address the comments.

---

## [Decision Letter · Decision Letter 2]

14 Feb 2022

PONE-D-21-09280R2Comparative efficacy of prophylactic anticonvulsant drugs following traumatic brain injury: a systematic review and network meta-analysis of randomized controlled trialsPLOS ONE

Dear Dr. Chen,

Thank you for submitting your manuscript to PLOS ONE. After careful consideration, we feel that it has merit but does not fully meet PLOS ONE’s publication criteria as it currently stands. Therefore, we invite you to submit a revised version of the manuscript that addresses the points raised during the review process.

We look forward to receiving your revised manuscript.

Kind regards,

Giuseppe Biagini, MD

Academic Editor

PLOS ONE

Journal Requirements:

Reviewers' comments:

Reviewer's Responses to Questions

**Comments to the Author**

1. If the authors have adequately addressed your comments raised in a previous round of review and you feel that this manuscript is now acceptable for publication, you may indicate that here to bypass the “Comments to the Author” section, enter your conflict of interest statement in the “Confidential to Editor” section, and submit your "Accept" recommendation.

Reviewer #1: All comments have been addressed

2. Is the manuscript technically sound, and do the data support the conclusions?

Reviewer #1: Partly

3. Has the statistical analysis been performed appropriately and rigorously? 

Reviewer #1: Yes

4. Have the authors made all data underlying the findings in their manuscript fully available?

Reviewer #1: Yes

5. Is the manuscript presented in an intelligible fashion and written in standard English?

Reviewer #1: Yes

6. Review Comments to the Author

Reviewer #1: The authors should be congratulated for providing an excellent revision, which has substantially improved the rigour of this meta-analysis. My concerns have been largely addressed by this revision. This is an interesting paper that provides useful data regarding the comparative efficacy of various antiepileptic medications as seizure prophylaxis in traumatic brain injury.

Major issues:

1. The authors have now added to the conclusion that they do not believe their findings support guidelines advocating the use of prophylactic phenytoin in traumatic brain injury. This is not supported by the data. Both the referenced Cochrane review, and the authors' own results, show that antiepileptics reduce the likelihood of early seizures. While there is currently no strong evidence to support any specific medication, phenytoin has the largest body of evidence for its efficacy and performed well in this meta-analysis despite the lack of statistical significance.

In addition, the efficacy of phenytoin relative to placebo IS arguably significant (OR 0.43, 95%CI 0.18-1.01). I would encourage the authors not to over-interpret p-values and the concept of 'statistical significance' in the context of network meta-analysis, where comparisons inevitably suffer from multiplicity (see doi: 10.1002/jrsm.1377). Therefore, the level IIA recommendation in the Brain Trauma Foundation guidelines to give one week seizure prophylaxis with phenytoin remains supported by the data in this study.

This is a good study but the conclusion in its current form is misleading and even potentially harmful because it implies that prophylaxis should not be given. The conclusion should be modified to make it clear that antiepileptic drugs reduce the risk of early seizures and the findings of this meta-analysis support current guidelines, but that the optimal medication is unclear. This can be considered a minor or major issue at the discretion of the editor, but should be rectified prior to publication.

Minor issues:

1. In the text, (page 9), the effect of phenytoin relative to placebo in reducing early seizures is described as OR 0.43, 95%CI 0.18-0.99. In the corresponding table S4, this is listed as OR 0.43, 95%CI 0.18-1.01. Please check these values and ensure they are correct and matching.

2. The results of the pairwise analysis comparing all AEDs versus placebo should also be included in the abstract, as the results of the paper cannot be properly interpreted without this knowledge.

7. PLOS authors have the option to publish the peer review history of their article (what does this mean?). If published, this will include your full peer review and any attached files.

Reviewer #1: **Yes: **Jack Henry

---

## [Author Response · Author response to Decision Letter 2]

23 Feb 2022

Thank you for your valuable comments that have significantly improved our manuscript. We have revised our manuscript based on your suggestions. The point-by-point responses to your specific comments are listed as below. We hope that our revisions are acceptable and that our responses adequately address the comments.

---

## [Decision Letter · Decision Letter 3]

11 Mar 2022

Comparative efficacy of prophylactic anticonvulsant drugs following traumatic brain injury: a systematic review and network meta-analysis of randomized controlled trials

PONE-D-21-09280R3

Dear Dr. Chen,

We’re pleased to inform you that your manuscript has been judged scientifically suitable for publication and will be formally accepted for publication once it meets all outstanding technical requirements.

Kind regards,

Giuseppe Biagini, MD

Academic Editor

PLOS ONE

Additional Editor Comments (optional):

Reviewers' comments:

Reviewer's Responses to Questions

**Comments to the Author**

1. If the authors have adequately addressed your comments raised in a previous round of review and you feel that this manuscript is now acceptable for publication, you may indicate that here to bypass the “Comments to the Author” section, enter your conflict of interest statement in the “Confidential to Editor” section, and submit your "Accept" recommendation.

Reviewer #1: All comments have been addressed

2. Is the manuscript technically sound, and do the data support the conclusions?

Reviewer #1: Yes

3. Has the statistical analysis been performed appropriately and rigorously? 

Reviewer #1: Yes

4. Have the authors made all data underlying the findings in their manuscript fully available?

Reviewer #1: Yes

5. Is the manuscript presented in an intelligible fashion and written in standard English?

Reviewer #1: Yes

6. Review Comments to the Author

Reviewer #1: The authors have addressed all of my concerns about this manuscript. This is an interesting meta-analysis that warrants publication.

7. PLOS authors have the option to publish the peer review history of their article (what does this mean?). If published, this will include your full peer review and any attached files.

Reviewer #1: No

---

## [Editor Report · Acceptance letter]

22 Mar 2022

PONE-D-21-09280R3 

Comparative efficacy of prophylactic anticonvulsant drugs following traumatic brain injury: a systematic review and network meta-analysis of randomized controlled trials 

Dear Dr. Chen:

I'm pleased to inform you that your manuscript has been deemed suitable for publication in PLOS ONE. Congratulations! Your manuscript is now with our production department. 

Kind regards, 

on behalf of

Dr. Giuseppe Biagini 

Academic Editor

PLOS ONE